# Heritability of Protein and Metabolite Biomarkers Associated with COVID-19 Severity: A Metabolomics and Proteomics Analysis

**DOI:** 10.3390/biom13010046

**Published:** 2022-12-27

**Authors:** Amelia K. Haj, Haytham Hasan, Thomas J. Raife

**Affiliations:** Department of Pathology and Laboratory Medicine, University of Wisconsin-Madison, 3170 UW Medical Foundation Centennial Building (MFCB), Madison, WI 53705-2281, USA

**Keywords:** COVID-19, SARS-CoV-2, biomarkers, metabolomics, proteomics, erythrocytes

## Abstract

Objectives: Prior studies have characterized protein and metabolite changes associated with SARS-CoV-2 infection; we hypothesized that these biomarkers may be part of heritable metabolic pathways in erythrocytes. Methods: Using a twin study of erythrocyte protein and metabolite levels, we describe the heritability of, and correlations among, previously identified biomarkers that correlate with COVID-19 severity. We used gene ontology and pathway enrichment analysis tools to identify pathways and biological processes enriched among these biomarkers. Results: Many COVID-19 biomarkers are highly heritable in erythrocytes. Among heritable metabolites downregulated in COVID-19, metabolites involved in amino acid metabolism and biosynthesis are enriched. Specific amino acid metabolism pathways (valine, leucine, and isoleucine biosynthesis; glycine, serine, and threonine metabolism; and arginine biosynthesis) are heritable in erythrocytes. Conclusions: Metabolic pathways downregulated in COVID-19, particularly amino acid biosynthesis and metabolism pathways, are heritable in erythrocytes. This finding suggests that a component of the variation in COVID-19 severity may be the result of phenotypic variation in heritable metabolic pathways; future studies will be necessary to determine whether individual variation in amino acid metabolism pathways correlates with heritable outcomes of COVID-19.

## Key Points

### **What is already known on this topic** 

Protein and metabolite changes are seen during SARS-CoV-2 infection, and plasma/serum biomarkers can be used to predict COVID-19 severity.

Erythrocyte proteomics and metabolomics data can be used to identify heritable metabolic pathways; heritable risk factors for COVID-19 severity have been previously identified, including host genetic variants associated with disease severity.

**The issue addressed by this study**: to help determine whether variation in COVID-19 severity may be explained in part by variation in heritable metabolic pathways.

## What This Study Adds

*The main take-home point of our study*: we find that COVID-19 biomarkers that are heritable in erythrocytes are enriched for metabolites involved in amino acid metabolism.

Our study suggests that a component of COVID-19 severity may depend on phenotypic variation in heritable amino acid metabolism pathways.

## 1. Introduction

As of December 2022, there have been 648 million cases worldwide of coronavirus disease 2019 (COVID-19), caused by severe acute respiratory syndrome coronavirus 2 (SARS-CoV-2), and over six million deaths. COVID-19 is notable for its wide range of clinical presentations, with most patients exhibiting mild disease, but with nearly 20% exhibiting symptoms ranging from hypoxia to respiratory failure and death [1]. Highly effective vaccines and boosters now offer reliable (yet imperfect) protection against infection and severe disease [2,3]. While factors such as age and the presence of comorbidities have been clearly linked to the likelihood of developing severe disease [4,5,6,7], less is known about the host molecular factors that underlie disease severity. Identification and characterization of molecular biomarkers of COVID-19 disease states may enable early identification of patients at increased risk for developing severe disease. Additionally, biomarkers may point to specific biological pathways that contribute to the development of symptomatic or severe disease.

Many studies have identified serum protein and metabolite biomarkers of COVID-19 in an effort to determine the molecular basis for severe disease [8,9,10,11,12]. These studies identify proteins and metabolites present at different concentrations in the serum of healthy individuals and individuals with varying degrees of COVID-19 severity. Identified biomarkers have pointed to disruptions in several biological pathways associated with COVID-19, including amino acid and lipid metabolism, and heme biosynthesis, among others. Several studies have also sought to identify alterations in erythrocytes as a result of COVID-19 infection, and have found a range of changes, including disruptions in levels of proteins involved in amino acid metabolism, changes in RBC morphology, and evidence of increased oxidative stress in RBCs [13,14,15]. 

In a previously performed classic twin study of erythrocyte proteins and metabolites, the authors identified over 700 unique molecules present in red blood cells and determined correlations among their levels as well as their heritability [16,17]. In addition to being easily collected, erythrocytes are metabolically active cells that maintain homeostasis in many metabolic pathways, enabling their use to identify relationships among proteins and metabolic pathways. Data from this twin study revealed heritability of entire metabolic pathways, including energy metabolism pathways [16]. Measurements of biomarker heritability, which frequently rely on twin studies but can also use studies of close relatives, offer a way to quantify the role of genetics in determining levels of specific proteins and metabolites. The heritability percentage of a biomarker is, therefore, the percentage of variation in its level that can be attributed to genetics rather than environmental factors [18]. While our previous twin study examined heritability of proteins and metabolites in the erythrocytes of healthy individuals, biomarker heritability has been studied in the context of numerous conditions, including chronic kidney disease, atherosclerosis, and metabolic syndrome [19,20,21,22].

We found that many molecules identified as heritable in our previous twin study overlap with those found in early, pre-vaccine COVID-19-omics studies. Here, we cross-reference two pre-vaccine studies of COVID-19 metabolomics and proteomics with the aforementioned twin dataset to identify heritable biomarkers and biological pathways associated with COVID-19 severity [8,9]. Understanding the heritability of known biomarkers of COVID-19 severity may not only explain whether some individuals are predisposed to severe disease, but also provide insights into SARS-CoV-2 pathogenesis.

## 2. Materials and Methods

### 2.1. Erythrocyte Multi-Omics Twin Study

This manuscript uses previously reported twin studies and represents data collected from the same study subjects [17,23,24,25]. The previously reported twin study was approved by the Human Subjects office of the University of Iowa Carver College of Medicine, and all participants provided written informed consent. As previously described, subjects all met the criteria for autologous blood donation according to the standard operating procedures of the University of Iowa DeGowin Blood Center. Health history and demographic information was obtained at the time of enrollment and informed consent. Twin pairs each donated one blood unit and did not necessarily donate concurrently. Five dizygotic and 13 monozygotic twin pairs participated in the study. Demographic information not required for blood donation was not collected.

Blood sampling and analysis and proteomics/metabolomics analysis were previously performed [16]. Briefly, whole venous blood collected from participants was centrifuged and the plasma and buffy removed. Aliquots of washed erythrocytes were lysed with nanopure water, mixed, and stored at −80 °C prior to proteomic and metabolomic analyses. DNA obtained from leukocyte reduction filters was used for zygosity testing. Samples were prepared as previously described for metabolomics analysis using ultrahigh performance liquid chromatography/tandem mass spectrometry (UHPLC-MS/MS) optimized for both basic and acidic species, and gas chromatography/mass spectrometry (GC-MS). Samples were also prepared as previously described for proteomics analysis using LC-MS/MS. Data analysis was performed as described using MaxQuant software version 1.5.2.8 [26] and the Andromeda search engine, and Pearson correlations were calculated among all proteins and metabolites using Perseus software [27]. Heritability calculations were performed using the one-way model of intraclass correlation coefficient (ICC), where ICC = (MS_between_ − MS_within_)/(MS_between_ + MS_within_) where MS_between_ is the mean-square variance estimate among all twin pairs and MS_within_ is the mean-square variance estimate within the group (dizygotic, DZ, or monozygotic, MZ). Heritability was estimated using the following equation: h^2^ = (ICC_MZ_ − ICC_DZ_)/(1 − ICC_DZ_).

### 2.2. Metabolite Pathway Enrichment Analysis

Metabolite pathway enrichment was characterized using the MetaboAnalyst 5.0 online Pathway Analysis tool (https://www.metaboanalyst.ca/MetaboAnalyst/home.xhtml, accessed 30 September 2020). Metabolite names were entered as a compound list, and when necessary, metabolite names were adjusted to match the nomenclature recognized by MetaboAnalyst. In three instances, no variation on a metabolite name was recognized by MetaboAnalyst, so these metabolites (4-vinylphenol sulfate, N-delta-acetylornithine, and succinylcarnitine) were excluded from analysis. Pathway analysis parameters used were as follows: Enrichment method: Hypergeometric Test, Topology analysis: Relative-betweenness Centrality, Reference metabolome: Use all compounds in the selected pathway library. The *Homo sapiens* KEGG pathway library was used as reference. Enriched pathways with FDR ≤ 0.05 were considered significant. 

### 2.3. Gene Ontology Analysis

Protein enrichment analysis was performed using the online Gene Ontology Resource (http://geneontology.org/, accessed 30 September 2020). Gene IDs were entered for biological process analysis, with adjustments to gene names made as necessary to ensure all were recognized. When multiple proteins were listed as a single item in either the COVID-19 datasets or the erythrocyte dataset, only overlapping proteins between the two datasets were used in Gene Ontology analyses. The analysis type used was the PANTHER Overrepresentation Test (released 20210224), the reference list used was Homo sapiens (all genes in database), test type used was Fisher’s Exact, and false discovery rate (FDR) correction was used. Both the “GO biological process complete” and “PANTHER Pathways” annotation datasets were used. Processes/pathways with FDR ≤ 0.05 were considered significant.

## 3. Results

### 3.1. Serum Biomarkers Associated with COVID-19 Disease States Overlap with Heritable Metabolites and Proteins in Erythrocytes

We first identified plasma biomarkers of COVID-19 disease states that exhibited heritability in RBCs using published datasets from [8,9]. Using the metabolomics data provided in [9], we identified 64 metabolites that were present at different levels across various COVID-19 disease severities that also demonstrated ≥30% heritability in erythrocytes (Table 1). Using proteomics data from both [8,9], we identified 13 differentially expressed proteins having ≥30% heritability in erythrocytes (Table 2).

### 3.2. Proteins Associated with COVID-19 Disease States Are Highly Correlated in Erythrocytes

To determine the relationships among proteins associated with COVID-19 disease states, we performed a Pearson correlation analysis using the erythrocyte protein expression data. For this analysis, we combined all proteins either up- or downregulated in COVID-19 disease states from both [8,9], regardless of their heritability in erythrocytes, and excluded proteins that are differentially expressed only in non-COVID disease states. Of the 14 upregulated proteins, 5.5% of the correlations had a Pearson correlation ≥ 75% (Figure 1), and of the 17 downregulated proteins, 11.8% of the correlations were ≥75% (Figure 2). The same analysis performed for the four metabolites upregulated in COVID-19 showed that none had correlations ≥ 75%, and for the 51 downregulated metabolites only 0.9% of the correlations were ≥ 75% (Appendix A). 

### 3.3. Decreased Levels of Heritable Metabolites Involved in Amino Acid Biosynthesis and Metabolism Pathways Are Associated with Both Non-Severe and Severe COVID-19

Previous analyses have found several biological pathways enriched among COVID-19 metabolite biomarkers. A meta-analysis by Pang et al. of several COVID-19-omics studies identified, among others, metabolites involved in porphyrin (heme) metabolism; valine, leucine, and isoleucine degradation; arachidonic acid metabolism; and unsaturated fatty acid biosynthesis pathways as being present at significantly different levels between COVID-19 patients and healthy controls [28]. Shen and colleagues, whose work was included in this meta-analysis, identified metabolites involved in amino acid metabolism, in particular arginine metabolism, as being significantly decreased in COVID-19 [9]. 

We sought to identify pathway associations among heritable proteins and metabolites to determine whether specific heritable pathways may be important in SARS-CoV-2 infection. The heritable metabolites upregulated in COVID-19 had no pathways significantly enriched. Among the metabolites downregulated in COVID-19 disease states that had ≥30% heritability in erythrocytes, amino acid metabolism and biosynthesis pathways were enriched. Specifically, enriched pathways with false discovery rates (FDR) ≤ 0.05 included valine, leucine, and isoleucine biosynthesis; glycine, serine, and threonine metabolism; and arginine biosynthesis, among others (Figure 3). This both corroborates previous findings and suggests that key metabolic pathways involved in COVID-19 may be heritable. Notably, among metabolites contained in the amino acid metabolism pathways that were also present in the erythrocyte dataset, the majority were heritable (Figure 4). Interestingly, however, the amino acids in question were not always themselves heritable in erythrocytes. Glycine, serine, and threonine were all ≥60% heritable, and arginine was 83% heritable, but leucine and isoleucine were ≤30% heritable, with only valine ≥ 30% heritable in that pathway. 

Among heritable proteins upregulated in COVID-19, no Gene Ontology terms were enriched; however, a PANTHER pathway analysis identified blood coagulation as an enriched pathway (FDR = 0.027). Among the heritable proteins downregulated in COVID-19, GO enriched terms were regulation of endothelial cell chemotaxis (FDR = 0.038) and positive regulation of epithelial cell migration (FDR = 0.025); however, no PANTHER pathways were enriched. Though these pathways are represented by smaller numbers of proteins compared to the metabolite enrichment analysis, these results again suggest that heritable biological processes may play a role in COVID-19.

## 4. Discussion

In this study, we interrogated a twin study database of erythrocyte proteins and metabolites to show that many of the plasma biomarkers associated with COVID-19 disease states are highly heritable in erythrocytes. The twin study previously showed inheritance of energy metabolism and redox pathways in erythrocytes, setting a precedent for its use to determine heritability of disease-associated pathways [16,17]. Metabolomics and proteomics studies of COVID-19 patients have exhaustively identified variations in protein and metabolite levels associated with the disease, but none have specifically addressed the heritability of biomarkers. Our results suggest that, in addition to individual biomarkers being heritable, several metabolic pathways downregulated in COVID-19, in particular amino acid biosynthesis and metabolism pathways, are heritable in erythrocytes. 

Understanding the heritability of disease biomarkers creates an opportunity to understand why certain individuals may be at heightened risk of severe disease and complications. To our knowledge, the finding that many COVID-19 biomarkers are heritable in erythrocytes is novel, as is the observation that many of these heritable biomarkers are part of amino acid metabolism pathways. These observations suggest that there may be familial risk of symptomatic or severe COVID-19 that warrants further exploration. Indeed, many studies have attempted to identify potential genetic risk factors for severe COVID-19. One study identified mutations in genes involved in type I interferon production as being associated with the development of severe pneumonia in the setting of COVID-19 [29]. A study of hospitalized Chinese patients found that those homozygous for a C at the rs12252 SNP in interferon-induced transmembrane protein 3 (IFITM3) gene (IFITM3) had more severe disease [30]. An analysis of UK Biobank data identified the ApoE e4e4 genotype, which is associated with dramatically increased risk of Alzheimer’s disease, as being associated with COVID-19 test positivity and severity [31]. Interestingly, a study using twin data found heritability of symptoms associated with COVID-19 [32]. The identification of many genetic variants that are associated with other respiratory illnesses suggests a potentially important role for genetics in COVID-19 susceptibility [33]. 

Our study suggests that amino acid metabolism pathways disrupted in COVID-19 [9] are heritable. In addition to the COVID-19-omics datasets our study referenced [8,9], many other studies have also described dysregulation of amino acid metabolism in COVID-19, although the directionality of these alterations in some cases differs among studies [34,35,36,37,38,39]. Amino acid metabolism is known to be required for immunity—beyond simply providing nutrients for protein synthesis, amino acids play a role in instructing cells to proliferate and carry out effector functions [40]. Levels of specific amino acids have been found to correlate with levels of proinflammatory cytokines [38], and amino acid metabolism has been linked to the development of cytokine release syndrome in COVID-19 [39]. A decrease in arginine is frequently seen among studies [9,34,37], and one study found that supplementation of arginine decreased the release of proinflammatory cytokines by peripheral blood mononuclear cells collected from SARS-CoV-2-infected macaques [39]. Taken together, these observations suggest that the dysregulation of amino acid metabolism pathways may be a targetable feature of severe COVID-19 infection. Our finding that these pathways are heritable suggests that genetics may govern, at least in part, the up- or downregulation of amino acid pathways that play a role in COVID-19 immune responses. 

We also found that aminoacyl-tRNA synthesis was a highly heritable pathway associated with COVID-19. Aminoacyl-tRNA synthetases have been found to be involved in immune cell development, and have also been associated with viral infections, in some cases being upregulated in response to infection and in others being specifically inhibited by viral RNA motifs [41]. The heritability of this pathway again points to a possible heritable impact on immunity that could influence COVID-19 severity. Additionally, other studies have identified changes in energy metabolism pathways in COVID-19 patients [13,38], which a previous study using our twin study dataset also found to be highly heritable in erythrocytes [16]. Unfortunately, many of the biomarkers identified as being predictive for COVID-19 severity were not in the twin erythrocyte data set, therefore we could not comment on heritability; however, it is possible that non-heritable biomarkers may present an opportunity to influence either likelihood of developing severe COVID-19 or alter the course of the illness. 

Limitations of this study include some aspects of the referenced datasets and the twin erythrocyte dataset. First, the COVID-19 datasets did not control for age, as noted by [9], so age may have acted as a confounder. Second, both COVID-19 datasets used plasma and serum for their metabolomics and proteomics experiments, while the twin multi-omics dataset used erythrocytes. To our knowledge, no systematic studies have been undertaken to demonstrate correlations between erythrocyte and plasma/serum protein and metabolite levels. One study published by Thomas et al. looked directly at protein and metabolite changes in the erythrocytes of COVID-19 patients. They found that while RBC parameters themselves were not significantly altered in COVID-19 patients, there were changes in energy and lipid metabolism pathways that pointed to changes in cell membrane homeostasis [13]. Analysis of the heritability of these specific changes in erythrocyte proteins and metabolites seen in COVID-19 patients using our erythrocyte-omics dataset is an important next step in evaluating the heritability of COVID-19 biomarkers. 

In spite of these limitations, we believe that our use of erythrocyte-omics data in this study provides valuable insight into the heritability of COVID-19 biomarkers. Erythrocytes enable detailed analyses of metabolic pathways in a viable cell, and our group and others have previously shown whole metabolic pathways to be heritable in erythrocytes [16,17,42]. A study of plasma metabolite heritability found that many of the amino acids whose metabolism pathways we identified as being heritable were themselves heritable in plasma [43]. With the exception of leucine and isoleucine, which were found to not be heritable in the twin erythrocyte data set (though their overall metabolism pathways were heritable) but were ≥30% heritable in plasma, amino acid heritability was higher in the erythrocyte dataset than in the plasma dataset. This may be reflective of the competing and complicating factors, such as kidney function and gut absorption, that impact serum and plasma more directly than they do erythrocytes. 

## 5. Conclusions

Taken together with the COVID-19 biomarker studies, our study shows that certain biomarkers of COVID-19 exhibit heritability, in particular those involved in amino acid metabolism and biosynthesis pathways. These pathways are instrumental in immune responses and have been found to be dysregulated in COVID-19. Our results are concordant with observations of heritable susceptibility to COVID-19. Further work will be required to determine whether variation in amino acid metabolism pathways correlates with heritable outcomes of COVID-19, and whether these associations are replicated in other tissue types, including erythrocytes from COVID-19 patients.

## Figures and Tables

**Figure 1 biomolecules-13-00046-f001:**
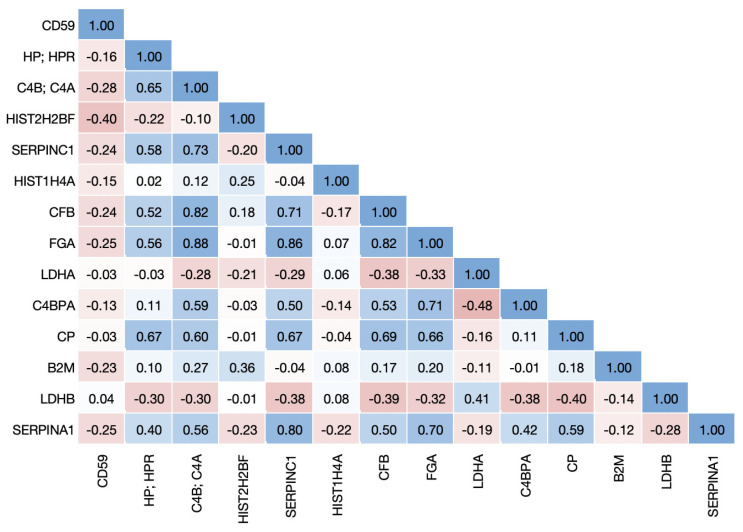
**Correlation matrix of proteins upregulated in COVID-19 disease states.** Proteins that were upregulated in non-severe and severe COVID-19 in the [9] dataset or upregulated in COVID-19 in the [8] dataset were identified in the erythrocyte dataset and Pearson correlations were calculated among all proteins.

**Figure 2 biomolecules-13-00046-f002:**
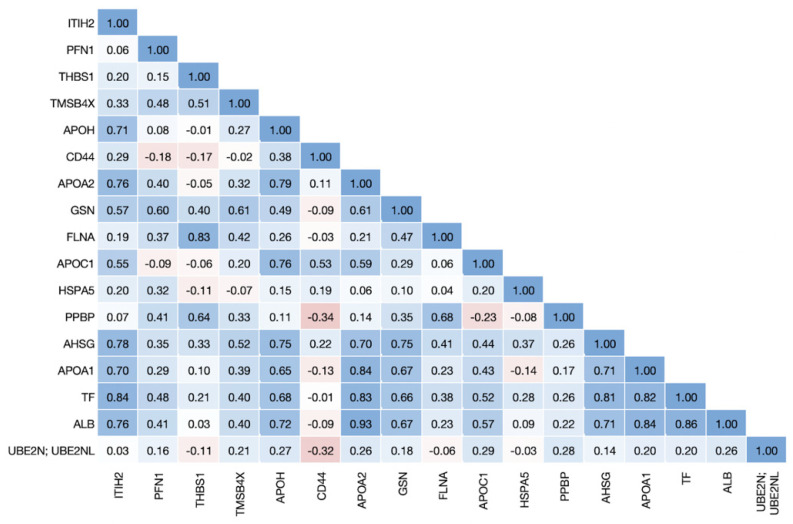
**Correlation matrix of proteins downregulated in COVID-19 disease states.** Proteins that were downregulated in non-severe and severe COVID-19 in the [9] dataset or downregulated in COVID-19 in the [8] dataset were identified in the erythrocyte dataset and Pearson correlations were calculated among all proteins.

**Figure 3 biomolecules-13-00046-f003:**
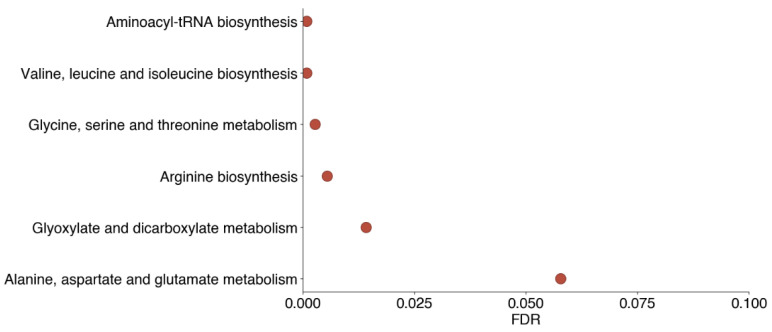
**Pathway enrichment among heritable metabolites downregulated in COVID-19.** Downregulated metabolites were identified from both [8,9], and metabolites with ≥30% heritability in erythrocytes were identified. Pathway enrichment was determined using MetaboAnalyst. Three metabolites (4-vinylphenol sulfate, N-delta-acetylornithine, and succinylcarnitine) were not available in the MetaboAnalyst database and were excluded from this analysis.

**Figure 4 biomolecules-13-00046-f004:**
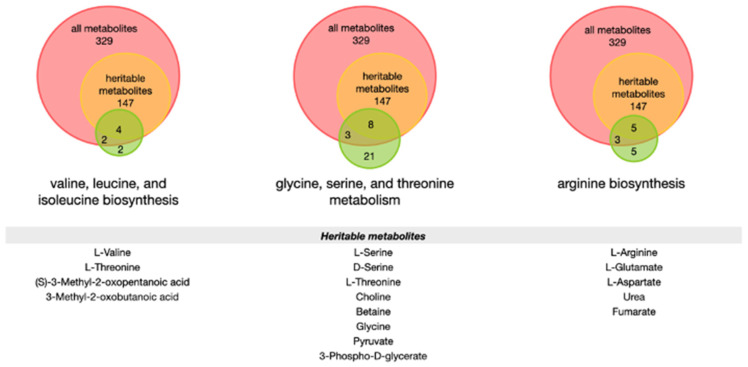
**Distribution of heritable vs. non-heritable metabolites involved in amino acid metabolism pathways.** Diagrams show the distribution of metabolites involved in valine, leucine, and isoleucine biosynthesis; glycine, serine, and threonine metabolism; and arginine biosynthesis compared to the metabolites present in the erythrocyte metabolomics dataset. Metabolites are classified as heritable (≥30% heritability) or non-heritable (“all metabolites”), or not present in the twin erythrocyte data set. Diagrams are not drawn to scale.

**Table 1 biomolecules-13-00046-t001:** Heritable metabolites differentially expressed in COVID-19 disease states. Metabolites with ≥30% heritability in erythrocytes were identified. ↑/blue indicates upregulation in the associated disease state (non-COVID-19, non-severe COVID-19, and severe COVID-19) in the [9] dataset; ↓/red indicates downregulation.

Metabolite Name	Heritability	Non-COVID vs. Healthy	Non-Severe vs. Healthy	Severe vs. Healthy	Severe vs. Non-Severe
glutamate	90.94		↓	↓	
thymol sulfate	90.01	↓	↓	↓	
betaine	87.62			↓	
cysteinylglycine	85.05	↓	↓	↓	
arginine	83.30		↓	↓	
spermidine	82.00	↑			
alpha-tocopherol	79.38	↓	↓	↓	
isobutyrylcarnitine	79.20		↓		
1-stearoylglycerophosphoserine	78.89	↓			
taurine	78.70		↓	↓	
adenosine	77.46		↓		
3-carboxy-4-methyl-5-propyl-2-furanpropanoate (CMPF)	73.68		↓		
gamma-glutamylglutamate	72.49		↓	↓	
threonate	72.35	↓	↓		
N-acetylmethionine	72.07		↓	↓	
methionine	71.34		↓	↓	
propionylcarnitine	71.26		↓		
fumarate	71.16	↑	↓		
threonylphenylalanine	71.08		↓		
13-HODE + 9-HODE	67.34	↑			
glycine	66.05		↓	↓	
isovalerylcarnitine	65.95	↑			
glycodeoxycholate	65.33	↓		↓	
leucylglycine	64.78		↓		
threonine	64.40		↓	↓	
1-arachidonoylglycerophosphocholine (20:4n6)	62.15			↓	
inosine	62.14			↑	
pyruvate	60.80		↓		
serine	60.07	↑	↓	↓	
4-vinylphenol sulfate	59.82	↓	↓	↓	
glycerol 3-phosphate (G3P)	59.03		↓	↓	
N-palmitoyl-D-erythro-sphingosine	58.24	↑	↑	↑	↑
choline	55.80		↓	↓	
epiandrosterone sulfate	55.53		↓		
urea	54.57		↓		
stearate (18:0)	54.02		↓	↓	
N-delta-acetylornithine	53.79		↓	↓	
docosapentaenoate (n6 DPA; 22:5n6)	52.87			↓	
1,5-anhydroglucitol (1,5-AG)	52.80			↓	
N-acetylmethionine sulfoxide	52.70	↑			
maltose	50.65	↑		↑	↑
alanine	50.60		↓	↓	
3-methylhistidine	50.17			↓	
hypotaurine	49.59		↓	↓	
phenylalanine	48.44	↑			
succinylcarnitine	48.24		↓		
valine	47.73		↓	↓	
urate	47.44			↓	
2-hydroxybutyrate (AHB)	47.31	↑		↑	
3-methyl-2-oxovalerate	45.36		↓	↓	
1-linoleoylglycerophosphocholine (18:2n6)	44.70	↓	↓	↓	
phenol sulfate	44.41	↓			
3-methyl-2-oxobutyrate	44.07		↓	↓	
margarate (17:0)	43.44		↓		
malate	43.35	↑	↓		
stachydrine	42.36		↓		
orotate	41.59	↑			
valylglycine	41.25		↓		
myo-inositol	38.95		↓		↑
oleate (18:1n9)	38.62	↑			
nicotinamide	38.10		↓		
N6-acetyllysine	37.20	↑	↓	↓	
3-indoxyl sulfate	36.66			↓	
cholesterol	35.88		↓		

**Table 2 biomolecules-13-00046-t002:** Heritable proteins differentially expressed in COVID-19 disease states. Proteins with ≥30% heritability in erythrocytes were identified. ↑/blue indicates upregulation in the associated disease state (non-COVID-19, non-severe COVID-19, and severe COVID-19) in the [9] dataset; ↓/red indicates downregulation. Proteins represented in the “Upregulated in COVID” column were identified in the [8] dataset; no downregulated proteins in this study met the heritability threshold.

Protein	Heritability	Non-COVID vs. Healthy	Non-Severe vs. Healthy	Severe vs. Healthy	Severe vs. Non-Severe	Upregulated in COVID
CD59	75.64			↑		
ITIH2	74.19	↓		↓	↓	
HP; HPR	59.44		↑	↑	↑	yes
CFH	57.00					yes
PFN1	52.35			↓		
C4B; C4A	51.68			↑		
THBS1	51.44	↓	↓	↓		
HIST2H2BF	51.29			↑		
SERPINC1	41.57		↑			
TMSB4X	40.56		↓			
FGG	40.46					yes
HIST1H4A	37.44			↑		
DEFA3; DEFA1	35.25	↑

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
