# Peer review of "Heritability of Protein and Metabolite Biomarkers Associated with COVID-19 Severity: A Metabolomics and Proteomics Analysis"

_biomolecules, 2022, doi:10.3390/biom13010046_

Round 1

Author Response

We greatly appreciate the time and care the reviewers took to provide their comments on our manuscript. We thank the reviewers for their generally positive comments and their constructive and thoughtful suggestions. Below we provide a point-by-point response to their specific concerns (indicated by bold text). Our changes in the manuscript are highlighted in yellow.

A couple of lines should be added to highlight the importance of mass vaccination as prophylactic strategy to control COVID-19.

We have added a mention of vaccination as an important strategy both for COVID-19 prevention and disease severity mitigation. Two supporting references are cited.

Line 52, only one reference is cited. More reference should be added.

We have added several additional references that identify risk factors associated with severity and mortality of COVID-19 infections.

Line 59-60: since the authors state that “Several studies identified serum protein and metabolite biomarkers of COVID-19.....” the references 3 and 4 are not enough.

We have added additional references.

Line 60-65: the authors should explicit that they used the datasets from reference 3 and 4 for their analyses and explain why they selected exactly those datasets.

Thank you for this comment; we have provided additional information regarding the use of the datasets in the introduction. We selected these datasets because there were few studies of COVID-19-omics early in the pandemic, and these datasets were exceptionally thorough. As these datasets were generated before COVID-19 vaccines were developed, they do not have the issue of vaccination status potentially confounding results, which we note in the text.

Line 68: a short explanation of the principle of proteins and metabolites hereditability should be provided, supported by appropriate references.

Thank you for identifying this as a weak point in the introduction – we have added a few sentences describing the concept of protein/metabolite heritability and the meaning of a heritability “percentage.” We also included several references supporting the use of heritability measurements in chronic diseases.

Line 89-90: the authors should rephrase as follows: “Blood sampling and proteomics/metabolomics analyses were previously performed [5].”

This change has been made.

Line 95-98: the authors should replace UHPLC/MS/MS with UHPLC-MS/MS, GC/MS with GC-MS and LC/MS with LC-MS/MS (without repeating the extended meaning of “LC/MS”).

These changes have been made.

Table 1 and Table 2:

- the use of “^” for upregulated and of “v” for downregulated is awful. Instead, as an example, the authors could use arrows like the following: ↑ ↓

- The cells’ color has not been defined in the table legends.

- Table 1: the meaning of asterisks next to metabolite names (e.g. 1- stearoylglycerophosphoserine) is not specified

- Table1: correct “Non-covid” with “Non-COVID”

- Table 2: the meaning of the dots in the cells of the last column (Upregulated in COVID) is not specified.

We thank the reviewers for their attention to detail. The tables now use arrows, and the table legends have been updated to reflect the change, and include the meaning of the cell colors (the colors are redundant with the arrows and are there solely for ease of visualization). Asterisks next to metabolite names were removed, and the dots in the “Upregulated in COVID” column have been changed to “yes” for clarity.

Line 153: correct “Non-covid” with “Non-COVID”

This change has been made.

Figure 4:

In addition to the distribution of metabolites involved in amino acid metabolism the authors should give a detail of the heritable ones (i.e. 4, 8 and 5 metabolites, respectively) of the selected amino acid metabolism pathways.

We have added a table beneath the existing figure showing the heritable metabolites in each amino acid metabolism pathway.

  1. Discussion:

Line 231: correct “Covid-19” with “COVID-19”

This change has been made.

The discussion on amino acid metabolism pathways is quite poor. The authors should improve it by comparing their data with several literature articles.

The following ones can be considered as milestones in the field and should be discussed and added in the references:

doi: 10.1038/s41467-021-21907-9

doi: 10.3390/ijms22179548

doi: 10.1016/j.metabol.2021.154739

doi: 10.1183/13993003.00284-2021

doi: 10.1038/s41598-021-85788-0

doi: 10.3389/fmicb.2021.723818

doi: 10.3390/ijms23052414

We thank the reviewer for the suggested references – we have cited most of them in a new section of the discussion in which we provide more detail on the amino acid metabolism changes seen in other COVID-19 studies. As there is a wide range of findings and associated hypotheses regarding their significance, some of them contradictory, we focus on arginine metabolism, which was seen in several studies to be downregulated and was found to be heritable in our study.

References 3, 4 and 5 should be cited without the writing “[dataset]”.

This change has been made.

Reviewer 2 Report

In their manuscript entitled "Heritability of protein and metabolite biomarkers associated with COVID-19 severity: A metabolomics and proteomics analysis" Haj et al. show that some of the altered metabolic/proteomic molecules in COVID-19 patients are also heritable in erythrocytes. This study provides important information regarding the plausible reason behind the different pathology/symptomatology of COVID-19 in distinct patients, in the light of the disease's relation to heritable traits. Nonetheless, in this Reviewer's opinion this manuscript needs some changes before its publication.

1) The introduction needs more citations, especially since some important and relative references are missing. For example, in the second paragraph of the Introduction section, only two references are mentioned, but it could be enriched by several more. I will hereby suggest some references, but the authors are of course free to add these or others they find more suitable. 10.1172/jci.insight.140327; 10.1021/acs.jproteome.0c00519; 10.3390/ijms21228623; 10.5937/jomb0-37514

2) Again in the Introduction section, in the third paragraph that focuses on erythrocytes, I find it crucial to add some information (and consequently add some publications) regarding the currently known changes in erythrocytes of COVID-19 patients. I hereby suggest some references, but again, the authors are of course free to add the references they find more suitable 10.3389/fphys.2021.825055;  10.3389/fphys.2022.849910; 10.1021/acs.jproteome.0c00606

3) By reading the Methods section, I do not completely understand if the authors used previously reported data or if they performed blood collection and consequent -omics analysis on the same subjects that were used in the referenced studies. Please clarify.

4) In their limitations paragraph, the authors mention that the COVID-19 datasets they used are in plasma or serum, while the twin multi-omics dataset used erythrocytes. It should be noted that there is a publication (10.1021/acs.jproteome.0c00606) that provides changes in the metabolome/proteome of RBCs and to my knowledge its dataset is available online. I understand that it would take a lot of time to include this dataset in their analysis, but the authors should at least mention this work, especially since it would be more proper to use a COVID-19 erythrocyte dataset.

Author Response

We greatly appreciate the time and care the reviewers took to provide their comments on our manuscript. We thank the reviewers for their generally positive comments and their constructive and thoughtful suggestions. Below we provide a point-by-point response to their specific concerns (indicated by bold text). Our changes in the manuscript are highlighted in yellow.

1) The introduction needs more citations, especially since some important and relative references are missing. For example, in the second paragraph of the Introduction section, only two references are mentioned, but it could be enriched by several more. I will hereby suggest some references, but the authors are of course free to add these or others they find more suitable. 10.1172/jci.insight.140327; 10.1021/acs.jproteome.0c00519; 10.3390/ijms21228623; 10.5937/jomb0-37514

Thank you for the excellent citation suggestions; we have included Kimhofer et al and Barberis et al in the introduction.

2) Again in the Introduction section, in the third paragraph that focuses on erythrocytes, I find it crucial to add some information (and consequently add some publications) regarding the currently known changes in erythrocytes of COVID-19 patients. I hereby suggest some references, but again, the authors are of course free to add the references they find more suitable 10.3389/fphys.2021.825055;  10.3389/fphys.2022.849910; 10.1021/acs.jproteome.0c00606

We thank the reviewer for these citation suggestions. The focus of our paper is primarily on the heritability of biomarkers associated with COVID-19 severity, and happens to use an erythrocyte dataset; making any claims about the effects of COVID-19 on erythrocytes themselves is beyond the scope of our present work. However, we appreciate that it does make sense to include a mention of erythrocyte studies in COVID-19, and we have included this in the second paragraph of the introduction, prior to introducing our study.

3) By reading the Methods section, I do not completely understand if the authors used previously reported data or if they performed blood collection and consequent -omics analysis on the same subjects that were used in the referenced studies. Please clarify.

We have updated the first paragraph of the methods section to clarify that the erythrocyte data used in this study was all previously collected and described.

4) In their limitations paragraph, the authors mention that the COVID-19 datasets they used are in plasma or serum, while the twin multi-omics dataset used erythrocytes. It should be noted that there is a publication (10.1021/acs.jproteome.0c00606) that provides changes in the metabolome/proteome of RBCs and to my knowledge its dataset is available online. I understand that it would take a lot of time to include this dataset in their analysis, but the authors should at least mention this work, especially since it would be more proper to use a COVID-19 erythrocyte dataset.

We greatly appreciate the reviewer bringing this publication to our attention. While the dataset of this paper is available and would be an excellent addition to our study, it unfortunately appears to be in a much more “raw” format than the selected datasets and would most likely require direct collaboration with the authors to transform the data into a format usable for our study. This would be an excellent next step for our study, though, and we have added this to the discussion in both the limitations section and as a future direction for this work.

Round 2

Reviewer 1 Report

I strongly recommend this paper for publication into Biomolecules journal.

Reviewer 2 Report

The authors have adequately addressed all my concerns, therefore I believe that the manuscript can be accepted in its present form.